Spatial and temporal shifts in the diet of the barnacle Amphibalanus eburneus within a subtropical estuary

Freeman Christopher J. 1 freemanc@si.edu
Janiak Dean S. 1
Mossop Malcolm 1
Osman Richard 2
http://orcid.org/0000-0002-4691-1569 Paul Valerie J. 1
1 Smithsonian Marine Station, Smithsonian Institution , Fort Pierce, FL , USA
2 Smithsonian Environmental Research Center , Edgewater, MD , USA
Forbes Valery
Electronic publication date: 2018 Aug 15
Publication date: 2018
Volume: 6
Electronic Location ID: e5485
Received 2018 May 11; Accepted 2018 Jul 30
Copyright: © 2018 Freeman et al.
Copyright year: 2018
Copyright holder: Freeman et al.
License: This is an open access article distributed under the terms of the Creative Commons Attribution License, which permits unrestricted use, distribution, reproduction and adaptation in any medium and for any purpose provided that it is properly attributed. For attribution, the original author(s), title, publication source (PeerJ) and either DOI or URL of the article must be cited.
License URL: https://creativecommons.org/licenses/by/4.0/

Keywords: Stable isotopes, Estuary, Indian River Lagoon, Suspension feeders, Barnacles, Generalist, Algal bloom

Funding: St. John’s River Water Management District Contract 27799 Smithsonian Marine Station Financial support for this project was provided by a grant from St. John’s River Water Management District (Contract 27799) and by the Smithsonian Marine Station. The funders had no role in study design, data collection and analysis, decision to publish, or preparation of the manuscript.

==============================
The success of many sessile invertebrates in marine benthic communities is linked to their ability to efficiently remove suspended organic matter from the surrounding water column. To investigate the diet of the barnacle Amphibalanus eburneus, a dominant suspension feeder within the Indian River Lagoon (IRL) of central Florida, we compared the stable isotopes ratios (δ13C and δ15N) of barnacle tissue to those of particulate organic matter (POM). Collections were carried out quarterly for a year from 29 permanent sites and at sites impacted by an Aureoumbra lagunensis bloom. δ13C and δ15N values of Amphibalanus eburneus varied across sites, but δ15N was more stable over time. There was a range of δ15N values of Amphibalanus eburneus tissue from 6.0‰ to 10.5‰ across sites. Because land-based sources such as sewage are generally enriched in 15N, this suggests a continuum of anthropogenic influence across sites in the IRL. Over 70% of the variation in δ15N values of Amphibalanus eburneus across sites was driven by the δ15N values of POM, supporting a generalist feeding strategy on available sources of suspended organic matter. The dominance of this generalist consumer in the IRL may be linked to its ability to consume spatially and temporally variable food resources derived from natural and anthropogenic sources, as well as Aureoumbra lagunensis cells. Generalist consumers such as Amphibalanus eburneus serve an important ecological role in this ecosystem and act as a sentinel species and recorder of local, site-specific isotopic baselines.

Introduction

Organisms that can exploit suspended organic matter fill a unique niche in aquatic ecosystems (Ricciardi & Bourget, 1999; Riisgård & Larsen, 2010). By efficiently consuming both living cells and detritus, suspension feeders play critical roles in the regulation of primary production and benthic-pelagic coupling of nutrients and organic matter (Gili & Coma, 1998). This feeding mechanism is widely successful in estuarine and coastal ecosystems, and coincides with highly diverse communities in benthic habitats (Karlson & Osman, 2012; Cresson, Ruitton & Harmelin-Vivien, 2016). Suspension feeders encounter various forms of organic matter that change over space and time (Richoux, Vermeulen & Froneman, 2014). These sources of nutrition vary in size and quality and can include bacteria [from <2 μm], phytoplankton [2–200 μm], zooplankton, and detritus from terrestrial plants, phytoplankton, and submerged aquatic vegetation (Deegan & Garritt, 1997; Hsieh et al., 2000; Cresson, Ruitton & Harmelin-Vivien, 2016).

Although suspension feeders appear to be ecologically similar and can occupy overlapping niches within a habitat, there is substantial variation in the feeding structures and mechanisms of coexisting species (Stuart & Klumpp, 1984; Lesser et al., 1992; Riisgård & Manríquez, 1997; Karlson, Gorokhova & Elmgren, 2015). This variation impacts an organism’s ability to acquire, sort, and select food particles and may allow individual species to specialize in a subset of available suspended particulate matter (Dubois et al., 2007a; Riisgård & Larsen, 2010; Dubois & Colombo, 2014; Richoux, Vermeulen & Froneman, 2014; Cresson, Ruitton & Harmelin-Vivien, 2016; Whalen & Stachowicz, 2017). For example, bivalves are able to sort and selectively feed on particles, releasing inorganic matter via pseudofeces and efficiently retaining high quality particles greater than 5 μm (Jørgensen, 1974; Møhlenberg & Riisgård, 1978; Riisgård, 1988; Galimany et al., 2017a, 2017b). Tunicates, barnacles, and gastropod mollusks, in contrast, largely lack structures that facilitate particle selection and are considered generalist or indiscriminate suspension feeders (Lesser et al., 1992; Petersen, 2007; Dubois et al., 2007a; Cresson, Ruitton & Harmelin-Vivien, 2016). These generalists are, however, capable of retaining a broader range of particles that may not be consumed by selective suspension feeders or even adopting an omnivorous nutritional strategy that allows them to feed at higher trophic levels (Lesser et al., 1992; Bone, Carre & Chang, 2003; Petersen, 2007; Decottignies et al., 2007; Beninger et al., 2007; Kach & Ward, 2008; Riisgård & Larsen, 2010; Richoux, Vermeulen & Froneman, 2014; Cresson, Ruitton & Harmelin-Vivien, 2016).

Coastal ecosystems are subject to seasonal shifts in environmental conditions and strong resource gradients from a combination of inputs from marine and terrestrial sources (Deegan & Garritt, 1997). In addition, food webs in these systems are increasingly impacted by anthropogenically-derived pollution and harmful algal blooms (Carlton, Newman & Pitombo, 2011; Lapointe et al., 2015; Galimany et al., 2017b). Divergence in feeding mechanisms among coexisting suspension feeders may lead to differential responses of species to these changes and shape community composition across sites (Dubois et al., 2007a; Cresson, Ruitton & Harmelin-Vivien, 2016). Trophic plasticity may allow generalist species to exploit a broader range of ecological niches and adapt to both natural and anthropogenic changes in food availability. Barnacles, for instance, are a dominant component of many intertidal and estuarine ecosystems and are one of the most prolific invaders into coastal ecosystems worldwide (Carlton, Newman & Pitombo, 2011). Their success is likely linked to their ability to consume a wide range of particle sizes from zooplankton to phytoplankton (down to 2-5 μm), detritus, and organic matter from both natural and anthropogenic sources (Barnes, 1959; Crisp & Southward, 1961; Lesser et al., 1992; Riisgård & Larsen, 2010; Dolenec et al., 2006). The diet of barnacles has also been shown to vary over space and time (Dolenec et al., 2006; Dubois et al., 2007a; Dubois & Colombo, 2014; Richoux, Vermeulen & Froneman, 2014) resulting from shifts in organic matter composition.

The Indian River Lagoon (IRL) of central Florida is a subtropical, shallow (mean depth of <1 m) estuary that spans 250 km of the east coast of central Florida. The IRL supports a high diversity of marine species due to a variety of habitats (mangrove, seagrass, oyster reefs, and artificial substrates) and its location in a tropical/temperate transition zone in close proximity to the Gulf Stream (Gilmore, 1995; Swain et al., 1995). Like many estuaries, both acute and chronic stressors are increasingly impacting communities within the IRL, leading to cascading effects throughout local food webs. For example, nutrient loading (Lapointe et al., 2015) and the loss of planktonic grazers have led to an increased frequency and severity of algal blooms (Phlips et al., 2014). The “superbloom” of a Picocyanobacteria and a Pedinophyceae (both 1-2 μm) sp. in 2011 and blooms of the brown tide Aureoumbra lagunensis (4-5 μm) in 2012, 2013, and the winter of 2016 (SJRWMD, 2013; Phlips et al., 2014; Kamerosky, Cho & Morris, 2015; Lapointe et al., 2015) were particularly devastating within the sublagoons of the northern IRL (NIRL). Although these algal species are all non-toxic, high concentrations of algal cells blocked sunlight from reaching seagrass beds and, ultimately, resulted in hypoxic events that led to fish kills within the NIRL (Phlips et al., 2014).

Surveys of epifauna communities at 90 sites in the NIRL have found high species diversity (175 taxa in 11 phyla) and a dominance of the barnacle Amphibalanus eburneus, with a mean percent cover of 30% to 40% across sites (D.S. Janiak, 2016, unpublished data; Fig. S1). The success of Amphibalanus eburneus across sites in the NIRL may be linked to its generalist feeding on available sources of suspended organic matter, but little is known about how the diet of Amphibalanus eburneus changes over time and space and whether these changes mirror general shifts in particulate organic matter (POM). To investigate this, we compared the stable isotope ratios (δ13C and δ15N) of Amphibalanus eburneus tissue to that of POM from the water column. Collections were carried out quarterly for a year and also during an Aureoumbra lagunensis bloom. We tested the following hypotheses: (1) the δ13C and δ15N values of Amphibalanus eburneus vary over space and time and will be closely tied to the δ13C and δ15N values of POM, and (2) Amphibalanus eburneus will demonstrate shifts in δ13C and/or δ15N values that indicate consumption of the brown tide Aureoumbra lagunensis.

Materials and Methods

Sample collection

As part of a project monitoring epifaunal community composition and diversity over space and time, we established 29 permanent monitoring sites spanning 150 km within the three sub-lagoons (Indian River and Mosquito Lagoons, and the Banana River) of the northern region of the greater IRL (NIRL; Fig. 1; Table S1). Collections of Amphibalanus eburneus (N = 5–10 individuals) were carried out at each of these monitoring sites on a quarterly basis (in January, April, July, and October of 2015). Permits for species collections were provided by the Florida Fish and Wildlife Conservation Commission (SAL-14-1567-SR). To test whether a bloom of Aureoumbra lagunensis influenced the diet of Amphibalanus eburneus, samples were also collected opportunistically at sites within the Banana River and IRL that were influenced by Aureoumbra lagunensis from December of 2015 to March of 2016 (Fig. S2; Galimany et al., 2017a). Barnacles were removed from mangrove prop roots or artificial substrates (dock and bridge pilings or seawalls) using a paint scraper and placed into a 4 L plastic bag containing seawater for transit back to the Smithsonian Marine Station. There were missing data for some sites on particular dates resulting from sample loss or inaccessibility.

Figure 1 Map of 29 permanent monitoring sites established in the three sublagoons (Banana River [B], Indian River Lagoon [I], and Mosquito Lagoon [M]) of the Northern IRL of central Florida (inset map).

Map data: Google, SIO, NOAA, U.S. Navy, NGA, GEBCO, and Landsat/Copernicus.

During seasonal and Aureoumbra lagunensis collections, a sample of seawater (20 L) for POM was also taken at each site to monitor the δ13C and δ15N values of general sources of particulate carbon and nitrogen available to Amphibalanus eburneus. In the laboratory, this seawater was prefiltered through 105 μm mesh and then filtered through a Millepore quartz fiber filter (2 μm porosity) stacked on top of a Whatman glass fiber (GF) filter (0.7 μm porosity) to obtain a single POM sample for each time point at each site. The organic matter content on GF filters was below detection limits, so POM δ13C and δ15N values are based on the results from organic matter on quartz fiber filters (2 μm porosity) that were stacked on top of the GF/F.

Sample preparation and stable isotope (δ13C and δ15N) analysis

Samples of Amphibalanus eburneus were kept separate and held overnight in flowing, sand-filtered seawater to allow for gut evacuation and then frozen at −20 °C. Once thawed, the shell diameter was measured for each individual (n = 10 per site for each sampling period) and, using forceps, all tissue within the shell of a barnacle was placed into an individual glass vial (Richoux, Vermeulen & Froneman, 2014). Tissue was dried at 60 °C for 24 h and homogenized using a mortar and pestle. Homogenized samples were acidified to remove carbonate by exposure to 12 N HCl fumes for 12 h, after which samples were returned to the oven at 60 °C for 24 h (Freeman & Thacker, 2011). Quartz and GF filters containing POM were also dried and acidified prior to analysis. POM was scraped from each filter, and POM from each filter and barnacle samples were separately weighed to the nearest 0.001 mg into tared tin capsules. Isotope analysis was carried out at the Stable Isotope Facility at UC Davis using a PDZ Europa ANCA-GSL (for barnacle tissue samples) or Micro Cube (for POM samples) elemental analyzer interfaced to a PDZ Europa 20–20 isotope ratio mass spectrometer (Sercon Ltd, Cheshire, UK). Isotope values are reported in δ notation in units of permille (‰).

Data analysis

Isotope values provide a time-integrated record of an organism’s diet, with δ13C values providing information about the primary sources of carbon fueling local food webs and δ15N values acting as a proxy for trophic level and the sources of nitrogen assimilated (Michener & Kaufman, 2007). To test the effect of season and site on the placement of Amphibalanus eburneus samples within bivariate (δ13C and δ15N) isotopic space, we calculated isotopic dissimilarity (measured as Euclidean distance) among samples and analyzed dissimilarity using a permutational multivariate analysis of variance (PERMANOVA) (Primer 6 with PERMANOVA+ add-on). Seasonal variation in individual isotope values was assessed using a Kruskal–Wallis One-Way Analysis of Variance (ANOVA). Linear regressions were used to investigate the relationship between the δ13C and δ15N values of Amphibalanus eburneus tissue and the δ13C and δ15N values of POM at each site. These analyses were carried out using Systat.

Results

There was significant variation in the δ15N and δ13C values of Amphibalanus eburneus tissue over space and time (PERMANOVA testing the effect of site [PseudoF29,1046 = 23.89; p(perm) = 0.001] and season [PseudoF3,1072 = 140.6; p(perm) = 0.001]) (Figs. 2A and 3A; Fig. S3 in supplemental information) during annual collections. Within each season, δ15N and δ13C values also varied among sites (PERMANOVA testing the effect of individual site nested within collection period: PseudoF96,976 = 56.58; p(perm) = 0.001) (Fig. S3). Annual mean δ15N values from each site ranged from 6.0‰ to 10.5‰, with depleted (6.0‰ to 7.7‰) δ15N values in the northern region of the IRL (site #s I1–5) and in the Mosquito Lagoon (site #s M1–6) compared to sites in the Banana River and the southern IRL (site #s B and I6–13) that had δ15N values between 8.3‰ and 10.5‰ (Fig. 2A). Annual mean δ13C values ranged from −22.3‰ to −17.8‰, with variation at finer spatial scales (within lagoons and even between geographically adjacent sites) than δ15N values (Fig. 3A; Fig. S3). In addition, δ13C values varied more across seasons than δ15N values (Kruskal–Wallis: H: 5063; p < 0.001 and H: 10.23, p < 0.05 for δ13C and δ15N, respectively) (Figs. 2A and 3A; Fig. S3).

Figure 2 Mean (±SE) δ15N values of Amphibalanus eburneus tissue (A) and particulate organic matter (POM; (B)) at individual sites within the three sublagoons of the northern Indian River Lagoon.

Sublagoons include the Banana River [B], Indian River Lagoon [I], and Mosquito Lagoon [M]. Data are shown as annual means (from each season from January to October of 2015) and mean (Amphibalanus eburneus tissue) and single POM values from an Aureoumbra lagunensis bloom in January 2016.

Figure 3 Mean (±SE) δ13C values of Amphibalanus eburneus tissue (A) and particulate organic matter (POM; (B)) at sites within the sublagoons of the northern Indian River Lagoon.

Sublagoons include the Banana River [B], Indian River Lagoon [I], and Mosquito Lagoon [M]. Data are shown as annual mean (from each season from January to October of 2015) and mean (Amphibalanus eburneus tissue) and single POM values from an Aureoumbra lagunensis bloom in January 2016.

The δ13C and δ15N values of POM varied across space and time (Figs. 2B, 3B and 4; Fig. S4), with a range of annual means from −24.6‰ to −20‰ and 3.3‰ to 8.2‰ for δ13C and δ15N, respectively. The average δ13C and δ15N values of POM at each site explained 22% and 71% of the variation in the mean δ13C and δ15N values of Amphibalanus eburneus at the same site (linear regression: r2 = 0.22; p < 0.01; N = 35 and r2 = 0.71; p < 0.001; N = 35 for δ13C and δ15N values, respectively) (Fig. 4; Fig. S4 for individual site values for each season). Tissue values of Amphibalanus eburneus were, on average, enriched in both δ15N and δ13C compared to POM (+2.78 ± 0.15 SE and +1.76 ± 0.17 SE for δ15N and δ13C, respectively) (Fig. 4; Fig. S4).

Figure 4 Mean (±SE) δ15N (A) and δ13C (B) values of Amphibalanus eburneus tissue at a site as a function of mean (±SE) δ15N and δ13C values of particulate organic matter (POM) at the same site.

Data include 29 sites within the sublagoons of the northern Indian River Lagoon. Tissue and POM samples were taken each season from January to October of 2015 (shaded circles) and also during an Aureoumbra lagunensis bloom in January of 2016 (open circles).

δ15N of Amphibalanus eburneus tissue was between 9.1‰ and 13.1‰ during the Aureoumbra lagunensis bloom, with an average enrichment of 1.8‰ (range of 0.7 to 2.7‰) compared to annual mean values (Figs. 2A, 4A and 5; Fig. S3). Likewise, δ13C of Amphibalanus eburneus tissue during the Aureoumbra bloom ranged from −21‰ to −19‰, with a mean enrichment of 0.9‰ (range of −0.2‰ to 2.1‰) compared to the annual mean at a site (Figs. 3A, 4B and 5; Fig. S3). The δ15N and δ13C values of POM were variable during the Aureoumbra lagunensis bloom, with enrichment at some, but not all, sites (mean enrichment of 0.3‰ [range of −1‰ to 1.6‰] and −0.4‰ [range of −4.6‰ to 2.7‰] for δ15N and δ13C, respectively) relative to the annual mean (Figs. 2B, 3B and 4; Fig. S4). Tissue values of Amphibalanus eburneus were enriched in both δ15N and δ13C relative to POM under bloom conditions (+4.37 ± 0.30 SE (range +3.47 to +5.9) and +1.91 ± 0.75 SE (range −1.59 to +4.83) for δ15N and δ13C, respectively (Fig. 4; Fig. S4).

Figure 5 Mean (±SE) δ15N and δ13C values of Amphibalanus eburneus for each season from January to October of 2015 and during an Aureoumbra lagunensis bloom in January 2016.

Values were calculated from δ15N and δ13C values of Amphibalanus eburneus from all sites within the three sublagoons of the Northern Indian River Lagoon.

Discussion

Trophic ecology of Amphibalanus eburneus

The δ13C and δ15N values of POM suggest that particulate sources of organic matter vary across seasons and small spatial scales in the IRL. Variation in δ13C values is likely due to inputs of organic carbon from a combination of marine phytoplankton (δ13C values of −18‰ to −24‰) and detritus from terrestrial C3 plants such as mangroves (−35‰ to −25‰) and seagrasses (−13.5‰ and −5.2‰) (Deegan & Garritt, 1997; Michener & Kaufman, 2007). Likewise, the δ15N values of POM provide information about the sources of nitrogen fueling local food webs. For instance, while depleted δ15N values suggest natural N-fixation, elevated δ15N values (>3‰) are suggestive of nitrogen derived from 15N-enriched sources such as wastewater (Lapointe et al., 2015). POM δ15N values in our study that range from ∼3 to 8‰ therefore suggest a continuum of impact from anthropogenically-derived nutrients across sites in the IRL. High levels of dissolved inorganic nitrogen and total dissolved nitrogen have been reported previously in regions of the northern IRL resulting from long water residence times and inputs of anthropogenically-derived nitrogen via surface water runoff and groundwater from septic tanks (Smith, 1993; Lapointe et al., 2015). Our data suggest that sites close to human development in the Banana River and southern sites in the Indian River (δ15N values of ∼6 to 8‰) are more impacted by these 15N-enriched sources than those in the more sparsely populated Mosquito Lagoon and northern IRL sites; our POM values are in general agreement with the magnitude of 15N enrichment found in macroalgae from this region of the IRL (Lapointe et al., 2015).

The isotopic composition of Amphibalanus eburneus tissue was coupled to the temporal and spatial dynamics of POM δ15N and, to a lesser extent, δ13C values, supporting the contention that barnacles are generalist suspension feeders utilizing predominant components of the organic matter pool in the water column (Cresson, Ruitton & Harmelin-Vivien, 2016). In contrast, if barnacles had a broader capacity to sort and select particles based on size or nutritional quality, we would expect less variation in the δ13C and δ15N of barnacle tissue over time and space and a decoupling of POM and Amphibalanus eburneus isotope values (Decottignies et al., 2007; Dubois et al., 2007a; Dubois & Colombo, 2014). For example, in the oyster Crassostrea gigas, the δ13C and δ15N values of POM explained less than 5% of the variation in isotope values of oyster muscle tissue (Marchais et al., 2013). Although this relationship was significant in our study (explaining 71% and 22% of the variance for mean δ15N and δ13C values at each site, respectively), the trend may have been even stronger if we had finer temporal resolution instead of a single POM isotope “snapshot” for each season from a site. The tissue of Amphibalanus eburneus was generally enriched in 13C and 15N (higher δ13C and δ15N values) relative to POM (by +1.76‰ and +2.78‰, respectively). Because consumers are generally enriched in both 13C and 15N due to the process of trophic enrichment, this implies the presence of a “trophic step” between particulate matter and Amphibalanus eburneus (Dubois et al., 2007b). The magnitude of trophic enrichment varies across species, but is generally hypothesized to range from <1.0‰ to 2.0‰ for δ13C and 3.0‰ to 3.6‰ for δ15N (DeNiro & Epstein, 1981; Zanden & Rasmussen, 2001; McCutchan et al., 2003; Dubois et al., 2007b). Our values are therefore well within the estimated range for trophic enrichment, supporting generalist feeding on bulk POM by Amphibalanus eburneus (Hsieh et al., 2000).

Unlike many suspension feeders, barnacles are able to consume higher trophic level prey such as zooplankton (Richoux, Vermeulen & Froneman, 2014). Enriched δ15N values of Amphibalanus eburneus from the IRL may therefore reflect feeding on zooplankton (Dix & Hanisak, 2015). We posit, however, that if zooplankton had been a dominant component of the diet of Amphibalanus eburneus in the IRL, we would have observed an additional trophic step between POM (predominately phytoplankton) and Amphibalanus eburneus tissue isotope values. Enriched δ15N values are thus likely the result of the passage of anthropogenically-derived nitrogen assimilated by phytoplankton into the epifaunal food web. Enriched δ15N values in POM and barnacle tissue at sites impacted by anthropogenically-derived nutrients has been reported before in other systems, with up to a 5‰ enrichment in barnacles from impacted sites and strong linear correlations between POM and barnacle δ15N values (Dolenec et al., 2006, 2007). Elevated δ15N values of POM and Amphibalanus eburneus tissue (>8‰) in the more urbanized regions of the NIRL (Banana River and southern sites in the Indian River) and relatively stable δ15N values across seasons therefore suggest chronic exposure to nitrogen from anthropogenic sources at some sites. In contrast, δ13C values within a site were more variable over both space and time, implying shifts in carbon sources over small spatial scales and the potential for seasonal fluctuations in phytoplankton productivity or growth rates (Cifuentes, Sharp & Fogel, 1988). It is possible that higher variability in δ13C values of Amphibalanus eburneus tissue over time may be due to differences in the turnover rate of this isotope relative to δ15N (Dubois et al., 2007b), but little is currently known about the turnover rate of carbon and nitrogen isotopes in barnacles.

Amphibalanus eburneus and algal blooms

The IRL has been exposed to multiple, acute algal blooms over the past decade, with a particularly detrimental superbloom in 2011 and recurring Aureoumbra lagunensis blooms in 2012, 2013, and 2016 (Phlips et al., 2014; Galimany et al., 2017a). The Aureoumbra bloom in 2016 occurred after we had gathered a year of baseline quarterly sampling on Amphibalanus eburneus feeding in the NIRL, providing an opportunity to investigate how the trophic ecology of Amphibalanus eburneus changes during an algal bloom. A single “snapshot” assessment of the δ15N and δ13C values of POM during the Aureoumbra bloom demonstrated δ15N values that were at or above annual means and δ13C values that were highly variable, with evidence of both enrichment and depletion in δ13C relative to annual means. In contrast, a more time integrated assessment (via δ15N and δ13C values of Amphibalanus eburneus tissue) revealed that δ15N and, to a lesser extent, δ13C values of Amphibalanus eburneus were enriched during the Aureoumbra bloom compared to annual means. For δ13C, this shift likely reflects a combination of source homogenization (predominately Aureoumbra) and isotopic fractionation associated with algal growth and high productivity that alters the δ13C signal at the base of the food web (Cifuentes, Sharp & Fogel, 1988). Although the average enrichment of δ13C in Amphibalanus eburneus tissue relative to POM was still within the range of a trophic step (mean 1.91‰ ± 0.75 SE) during the bloom, substantial variation in this enrichment (range −1.59‰ to 4.83‰) suggests that direct reliance of Amphibalanus eburneus on phytoplankton may be reduced during blooms.

Enrichment of δ15N (up to 13‰) may reflect increased Amphibalanus eburneus consumption of phytoplankton that are relying on enriched sources of anthropogenically-derived N. Although blooms of Aureoumbra in the IRL have previously been shown to elicit lower POM15N values than under non-bloom conditions (Kang, Koch & Gobler, 2015), Aureoumbra is also known to rapidly assimilate NH4, a common component of nitrogen derived from anthropogenic sources such as septic tanks (Lapointe et al., 2015; Kang, Koch & Gobler, 2015). We therefore propose that Amphibalanus eburneus may be consuming 15N-enriched Aureoumbra cells at bloom sites. Alternatively, because barnacles have been shown to reduce feeding efficiency at lower particle sizes (3 to 5 μm; Lesser et al., 1992), enriched δ15N values under bloom conditions may also reflect an additional trophic step (δ15N values of Amphibalanus eburneus tissue was on average enriched by 4.37‰ (range of 3.47‰ to 5.9‰) compared to POM during the bloom) as Amphibalanus eburneus is feeding more heavily on zooplankton than phytoplankton during an algal bloom. These data provide initial evidence of nutritional shifts in Amphibalanus eburneus under bloom conditions, but additional work in the laboratory is needed to verify the role of this epifaunal species in bloom mitigation (Galimany et al., 2017b).

Conclusions

The generalist feeding strategy of Amphibalanus eburneus appears to allow it to exploit spatially and temporally variable sources of organic matter and may contribute to the successful dominance of this species across sites in the NIRL (Carlton, Newman & Pitombo, 2011; Karlson & Osman, 2012). The abundance of Amphibalanus eburneus on diverse substrates and across both impacted and pristine sites within the NIRL is in direct contrast to other epifaunal organisms that are more constrained in their distribution or are currently present at only a fraction of their historical abundance (e.g., Crassostrea virginica and Mercenaria mercenaria; MacKenzie, Taylor & Arnold, 2001; Wilson et al., 2005; Garvis, Sacks & Walters, 2015; D.S. Janiak, 2016, unpublished data). As a dominant and stable faunal component of the NIRL, Amphibalanus eburneus is likely playing an important role in nutrient and organic matter cycling in this system (Dubois et al., 2007a). In addition, with a wide distribution and an integration of diverse sources of carbon and nitrogen into its tissues, Amphibalanus eburneus acts as an important sentinel species and recorder of isotopic baselines (Post, 2002; Dolenec et al., 2006; Fukumori et al., 2008). Finally, its nutritional plasticity may also allow Amphibalanus eburneus to capture and consume bloom particles such as Aureoumbra, providing a potential means for bioremediation and the prevention of algal blooms via top-down control.

Supplemental Information

Supplemental Information 1 Raw data from study.

Click here for additional data file.

Supplemental Information 2 Table S1. Names (with abbreviation), habitat information, and GPS coordinates for the 29 permanent monitoring sites established within the greater North Indian River Lagoon.

Click here for additional data file.

Supplemental Information 3 Fig. S1. Mean (+/−SE) percent cover of six most common epifaunal species or functional groups during initial (top) and final (bottom) surveys of epifaunal habitats within the Indian River Lagoon.

The same 90 sites were surveyed both times.

Click here for additional data file.

Supplemental Information 4 Fig. S2. Mean (+/−SE) chlorophyll a concentration (μg/L) in the water column at three sites of the NIRL over a two-year period including the Aureoumbra lagunensis bloom starting in December of 2015.

Data were derived from St. Johns River Water Management District: http://webapub.sjrwmd.com/agws10/hdswq/.

Click here for additional data file.

Supplemental Information 5 Fig. S3. Mean (+/−SE) δ15N (a) and δ13C (b) values of Amphibalanus eburneus tissue at 29 sites.

Sites were within the sub-lagoons of the North Indian River Lagoon (Banana River [B], Indian River Lagoon [I], and Mosquito Lagoon [M]). Data are shown for each season from January to October of 2015 and during an Aureoumbra lagunensis bloom in the January of 2016.

Click here for additional data file.

Supplemental Information 6 Fig. S4. Mean δ15N (a) and δ13C (b) values of Amphibalanus eburneus tissue at a site as a function of the δ15N and δ13C value of particulate organic matter (POM) at the same site.

Data include 29 sites within the sublagoons of the North Indian River Lagoon. Tissue and POM samples were taken each season from January to October of 2015 and during an Aureoumbra lagunensis bloom in the January of 2016.

Click here for additional data file.

We thank the Smithsonian Marine Station staff and, in particular, Sherry Reed and Woody Lee for their field assistance. Lab assistance was provided by A. Domingos, J. Houk, and J. Reyes. This is contribution number 1095 from the Smithsonian Marine Station.

Additional Information and Declarations

Competing Interests

Author Contributions

Field Study Permissions

Data Availability

The authors declare that they have no competing interests.

Christopher J. Freeman conceived and designed the experiments, performed the experiments, analyzed the data, contributed reagents/materials/analysis tools, prepared figures and/or tables, authored or reviewed drafts of the paper, approved the final draft.

Dean S. Janiak conceived and designed the experiments, performed the experiments, analyzed the data, prepared figures and/or tables, authored or reviewed drafts of the paper, approved the final draft.

Malcolm Mossop performed the experiments, analyzed the data, authored or reviewed drafts of the paper, approved the final draft.

Richard Osman conceived and designed the experiments, performed the experiments, authored or reviewed drafts of the paper, approved the final draft.

Valerie J. Paul conceived and designed the experiments, performed the experiments, contributed reagents/materials/analysis tools, authored or reviewed drafts of the paper, approved the final draft.

The following information was supplied relating to field study approvals (i.e., approving body and any reference numbers):

Permits for species collections were provided by the Florida Fish and Wildlife Conservation Commission (SAL-14-1567-SR).

The following information was supplied regarding data availability:

The raw data are provided in a Supplemental File.

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
