# Peer review of "Spatial and temporal shifts in the diet of the barnacle Amphibalanus eburneus within a subtropical estuary"

_PeerJ, doi:10.7717/peerj.5485_

## Round 0.1 · original submission · Minor Revisions

Please return a marked revision as well as a separate letter indicating how you have responded to each of the reviewers' comments.

Reviewer 1 ·

Basic reporting

The authors present a clearly written, well-thought out manuscript on the generalist feeding strategy of barnacles across a broad spatial scale over the course of a year. Overall, the manuscript reads very well, although there are a few spots which I think could receive a little more attention.

LN 87-90: Can you provide a range (if available) of particle sizes that barnacles can effectively remove from the water column?
Ln 97-99: I think there is a word missing in this sentence before location?
LN 103-107: What was particularly devastating about these blooms? Are they toxic? What is the size of the brown tide cells? It would be beneficial to provide a little more details on the bloom species.
LN 238: Does decoupling occur during the brown tide blooms? It looks like it might.
LN 242: I am curious about the C signature in the tissues here (and I have also brought this up below) - is there any literature on barnacle tissue turnover time? isotopic signatures are integrated over time, and while the N may stay more stable if the sources of N are chronic, I wonder if the variability in the C isotopes has anything to do with tissue turnover/growth rates. Thoughts?
LN 288-290: Is this enrichment also possibly indicative of a second trophic step, perhaps that during bloom conditions, the barnacles do feed on zooplankton? Is it possible that despite the generalist feeding nature of barnacles that the brown tide is too small to be captured by their filter?

Experimental design

The authors use a sound research approach, particularly as it pertains to hypothesis 1 - that isotopic values vary over space and time and will be closely linked to POM. I am less convinced of the approach to hypothesis 2 in effectively showing that the barnacles are consuming the brown tide plankton. I don't think this is a major issue, just one that should be discussed further - based on the isotope data presented alone, it is not clear that the barnacles consume the brown tide. Experiments would have been more useful in showing this.
Additionally, a little more detail on the analysis - particularly, it is not clear if bloom samples were included in the regression analysis (also unclear if they should be), or whether the bloom samples were analyzed separately (which would at least add more teeth to the argument about barnacles consuming brown tide). A little clarification here would be beneficial.

Validity of the findings

As previously mentioned, this was a sound research approach, and particularly the temporal and spatial exploration of barnacle tissue and POM isotopic values is robust and technically sound. The authors were able to demonstrate high spatial and temporal variability in isotopic signatures of POM and barnacle tissues, and despite this variability, the two tracked each other very well, especially during non-bloom periods. This was especially true for nitrogen, but less so for carbon. I have a few questions regarding this - there is considerably more variability in the C signatures. Do the authors have any ideas as to why? This could receive a little more attention in the discussion. Additionally, is there any information about the tissue turnover times of barnacles in general, or this species in particular? The authors present compelling evidence regarding the fairly steady N conditions in the IRL, but the C source likely changes quite a bit over time, and the isotopic signature of tissues is reflective of not only the source of C, but the source integrated over some period of time, and so there might be a lag in the C signature of the POM and the tissues. Can this be expanded upon?

The effects during the bloom are a little less convincing - yes there is changes in the isotopic signatures of tissues compared to the annual mean, but without specific analysis of this, I am not sure the conclusions are very well supported by the data. I think there is an interesting story here - and there does appear to be a shift in signature during these bloom periods, but I am not sure if this is because they are consuming the brown tide or if there is some selectivity during this time period. Could the authors do analysis of just the brown tide samples similar to the other POM samples to show that the tissue still tracks the POM during the brown tide? Further, the N signatures during the brown tide are enriched 6-7 permille, which might be more indicative of a 2 step trophic shift. This should at least be discussed as a possibility.

Additional comments

I found this manuscript to be well-written and thorough. I feel the issues I have presented here should be very easy to address in a minor revision. I commend the authors on their good work.

·

Basic reporting

This manuscript is very well written, and the authors represent previous literature well within their Introduction. The article is structured appropriately and the raw data is shared. The hypotheses are clearly stated and the results are relevant to the hypotheses.

Experimental design

The study at hand is a sound investigation of barnacle diet, focusing on the seasonal variability in these diets. The researchers’ additional interest in anthropogenic effects, in the form of algal blooms, further contributes to the scientific understanding of human impacts on coastal species and the ability of those species to adapt to perturbations.

The study question and hypotheses are well defined by the authors. However, the justification provided for the question is slightly mismatched from the study design. To truly address the study goal [(ll. 109-112) “To determine whether the abundance of A. eburneus across sites in the NIRL is linked to generalist feeding on available sources of suspended organic matter…”], the researchers should have not only assessed the diets of Amphibalanus eburneus, but also the diets of the other less dominant suspension feeders in the community. This can be remedied by simply rewording the study goal.

Both field collections and laboratory procedures were conducted appropriately and with the necessary permits. Methods are described in sufficient detail for replication.

Validity of the findings

The seasonal representation of POM values in IRL is a valuable contribution to the ecological understanding of the area. It also provides information about the level of human impact on IRL sites.

The evaluated coupling of POM and A. eburneus tissue values is warranted and conclusions are valid given the reported results.

The authors’ conclusions breakdown a bit when they begin to justify zooplankton as a lesser component of A. eburneus diets. Though I agree that the authors would have seen an additional trophic step between POM and A. eburneus tissue if the barnacles were consuming large quantities of zooplankton, I did not follow the authors’ logic about body size (ll. 258-261). If the authors are positing that larger body size correlates with feeding at a higher trophic level in barnacles, references to previous literature showing as much needs to be included. The following line of evidence noted – the strong correlation between 15N of A. eburneus and salinity – is similarly diffuse and requires more explanation and reference to the literature (you do this well in the Abstract ll. 32-34). The statements presented in ll. 264-273 present much clearer evidence for the validity of the authors’ findings.

The Conclusion contains statements that link the findings of the present study to a conclusion that, “The generalist feeding strategy of A. eburneus...likely contributes to the successful dominance of this species across sites in the NIR,” (ll. 300-302). Though this may be true, I do not think the authors can make this claim as a central conclusion given that their methods did fully address the question backing this statement (see my comments in Experimental Design). This point can still be made, but it should not be considered “likely”.

Additional comments

In addition to addressing the points raised in the previous sections, please address the edits listed below:

ll. 107-109 and Figure S1 – Why are amphipods included in this? Are they tube dwelling amphipods that create substrate? Additionally, the categories are rather different between 2014 and 2016 – were the same sites used? It’s ok if they’re not, but the authors should state this clearly. Finally, you say “suspension feeders”, but you have included algae – please justify.
l. 130 – Start a new paragraph with “To test whether a bloom…”
l. 346 – Add a comma after “Deegan”.
ll. 359-363 – Switch the position of these two citations.
l. 392 – Add a period after the pages.
ll. 393-394 – Title of the article should not be in caps for all words.
l. 411 – Add comma after MacKenzie.
l. 412 – “eds.” Needs to be consistently capitalized or lower case throughout citations.
l. 419 – Add comma after Michener.
l. 426 – Add comma after Phlips.
l. 429 – Remove comma after Coasts.
l. 430 – You previously capitalized the word after the semicolon in l. 428. Keep it consistent.
l. 431 – Remove comma after Ecology.
l. 440 – Keep capitalization after semicolons consistent throughout citations.

Figure 5 caption – Add period after “clarity”.
Figure S4 – I think the manuscript would benefit from this figure being moved to the main text. This gives the reader a much more visually succinct understanding of how A. eburneus tissue values change with season and algal bloom.

---

## Round 0.2 · accepted · Accept

The revised version has adequately addressed the reviewers' comments, and I find the paper acceptable for publication.

#